# AI-Powered Analysis of Weight Loss Reports from Reddit: Unlocking Social Media’s Potential in Dietary Assessment

**DOI:** 10.3390/nu17050818

**Published:** 2025-02-27

**Authors:** Efstathios Kaloudis, Victoria Kouti, Foteini-Maria Triantafillou, Patroklos Ventouris, Rafail Pavlidis, Vasiliki Bountziouka

**Affiliations:** 1Computer Simulation, Genomics and Data Analysis Laboratory, Department of Food Science and Nutrition, School of the Environment, University of the Aegean, 81400 Myrina, Greece; stathiskaloudis@aegean.gr (E.K.);; 2Population, Policy and Practice Research and Teaching Department, GOS Institute of Child Health, University College London, London WC1N 1EH, UK; 3Department of Cardiovascular Sciences, University of Leicester, BHF Cardiovascular Research Centre, Glenfield Hospital, Groby Road, Leicester LE3 9QP, UK

**Keywords:** ketogenic diet, large language models (LLM), artificial intelligence (AI), dietary assessment, weight loss, social media analytics, self-reported data

## Abstract

**Background/Objectives**: The increasing use of social media for sharing health and diet experiences presents new opportunities for nutritional research and dietary assessment. Large language models (LLMs) and artificial intelligence (AI) offer innovative approaches to analyzing self-reported data from online communities. This study explores weight loss experiences associated with the ketogenic diet (KD) using user-generated content from Reddit, aiming to identify trends and potential biases in self-reported outcomes. **Methods**: A dataset of 35,079 Reddit posts related to KD was collected and processed. Posts mentioning weight loss, diet duration, and additional factors (age, gender, physical activity, health conditions) were identified, yielding 2416 complete cases. Descriptive statistics summarized weight loss distributions and diet adherence patterns, while linear regression models examined factors associated with weight loss. **Results**: The median reported weight loss was 10.9 kg (IQR: 4.4–22.7 kg). Diet adherence varied with 36.3% of users following KD for up to 30 days and 7.8% for more than a year. Metabolic (27%) and cardiovascular disorders (17%) were the most frequently reported health conditions. Adherence beyond one year was associated with an average weight loss of 28.2 kg (95% CI: 25.5–30.9) compared to up to 30 days. Male gender was associated with an additional weight loss of 5.2 kg (95% CI: 3.8–6.6) compared to females. **Conclusions**: Findings suggest KD may lead to substantial weight loss based on self-reported online data. This study highlights the value of social media data in nutritional research, uncovering hidden dietary patterns that could inform public health strategies and personalized nutrition plans.

## 1. Introduction

The integration of artificial intelligence (AI) into healthcare is transforming the landscape of clinical practice, offering innovative solutions to enhance efficiency, accuracy, and personalization [1,2]. In dietary assessment, AI-powered tools provide the ability to analyze vast datasets, automate food tracking, and generate personalized dietary recommendations [3,4]. By leveraging machine learning algorithms, natural language processing (NLP), and computer vision, AI enables clinicians to gain deeper insights into dietary habits and their impact on health outcomes [5,6]. This technological evolution is particularly significant given the limitations of traditional dietary assessment methods, such as food recalls and self-reported logs, which are often prone to inaccuracies and underreporting [7].

Social media platforms and online forums have emerged as valuable sources of information and data in modern dietary research [8,9,10]. Platforms such as Twitter, Instagram, and Reddit host extensive discussions on dietary habits, personal experiences, and health outcomes. These platforms not only shape dietary trends but also serve as repositories of unfiltered user-generated content, providing a unique opportunity to study real-world dietary practices [11]. However, the widespread sharing of dietary advice by influencers, untrained users, and self-proclaimed nutrition experts raises concerns about the alignment of such recommendations with official dietary guidelines and scientific consensus. Studies have shown that much of the nutrition-related content circulating on social media lacks scientific rigor and often contradicts established dietary recommendations [12]. This discrepancy underscores the need for a systematic analysis of social media data to assess the validity of self-reported dietary experiences and their implications for public health [13].

Among social media platforms, Reddit stands out as a particularly useful source for dietary assessment due to its highly engaged user base with over 97 million daily active users and more than 100,000 diverse communities (i.e., “subreddits”) [14]. Reddit communities dedicated to health and nutrition, such as “r/keto”, “r/nutrition”, and “r/loseit”, offer a wealth of qualitative and quantitative data on dietary habits and health outcomes. The ketogenic diet (KD), characterized by high fat, moderate protein, and low carbohydrate intake [15], has garnered significant attention both in clinical research and popular media. Its health implications, particularly in promoting weight loss and improving metabolic health, have been widely studied [16,17,18]. On Reddit, the ketogenic diet (“r/keto”) ranks among the top 1% of all subreddits, with around 4 million users, making it the most prominent dietary pattern discussed on the platform. Users share their personal experiences, progress updates, and challenges associated with KD, making Reddit an ideal platform for examining its effects on weight loss.

Given the potential of AI to analyze unstructured data and extract meaningful patterns, this study aims to investigate the relationship between ketogenic diet and reported weight loss outcomes as discussed in Reddit. By leveraging AI tools to analyze Reddit data, this study seeks to provide insights into how real-world adherence to the ketogenic diet correlates with weight loss, while also exploring the broader application of AI in dietary assessment. Specifically, we aim to demonstrate the feasibility and utility of using AI-powered methods to analyze social media data for dietary research with a focus on the ketogenic diet as a case study.

## 2. Materials and Methods

In this study, we selected Reddit as the primary data source due to its extensive user base and the richness of its user-generated content, which spans diverse topics, including health and diet. Among the numerous communities (subreddits) dedicated to dietary discussions, we chose “r/keto” for the purposes of this study. This subreddit, created on 27 May 2010, is the most popular community centered on the ketogenic diet. It has over 3.9 million members, ranked within the top 1% of all subreddits, and placed at the 200th position in overall subreddit popularity. As the leading subreddit dedicated to diet-related topics, it is an ideal dataset for studying real-world dietary practices and outcomes. To obtain the data, we utilized publicly available torrent files hosted by the subreddit “r/pushshift”, which provides comprehensive archives of Reddit content [19]. The dataset used in this study consists of anonymized Reddit posts from the “r/keto” subreddit [20].

These datasets include both original submissions and comments. However, for the purposes of this study, we focused exclusively on analyzing the original submissions, as they contain more structured and self-contained information relevant to our research questions.

To extract and analyze the textual content of each post and answer a predefined set of questions, we employed the Llama-3-8B-Instruct model from Hugging Face [21], which is a state-of-the-art large language model (LLM). The implementation and manipulation of the data and the Llama model were conducted using the Python programming language (version 3.12.3). Python was chosen for its flexibility and extensive libraries that facilitate both natural language processing and data analysis. The model was tasked with extracting key numerical and categorical self-reported information from each submission, including weight loss, diet duration, gender, age, height, physical activity status, and any health conditions mentioned. The prompt used to guide the model’s output is shown in Table 1.

To ensure refined and structured data for subsequent analysis, we utilized regular expressions (regex) in Python to standardize the extracted information. For instance, all reported weights were converted to kilograms, and durations were normalized to days. Regex expressions allowed us to handle the variability and inconsistencies in user-generated content, ensuring uniformity in the dataset. To identify posts with highly similar textual content (duplicates), the Python library TheFuzz [22] was applied, which uses the Levenshtein distance to measure the similarity between text strings. This method allows for the detection of near-duplicates even in the presence of minor changes (e.g., typos, paraphrasing, or formatting changes). The code for data extraction and processing was executed on a Dell G15 5530 laptop (Dell Inc., Austin, TX, USA) equipped with 16 gigabytes of RAM and an Intel^®^ Core™ i7-13650HX processor (Intel Inc., Santa Clara, CA, USA) with 14 total cores.

We have randomly selected approximately 1% of the available posts (*n* = 838) to check the accuracy of the LLM. Four independent reviewers (VK, RP, MFT, PV) compared the results returned by the LLM with the original posts. The cumulative accuracy score across the reviewed sample for all domains is shown in Figure 1. As can be seen, accuracy stabilizes after approximately 300 posts, despite the initial variability, and remains consistent as additional cases are reviewed. The pattern is similar for all parameters tests, and it suggests that validating more than 1% of the posts would not significantly change the accuracy estimates.

Overall, the correct classification rate ranged from 72% for weight loss and diet duration to 90–95% for age, gender, height, physical activity and health condition. The LLM was tasked with identifying seven domains (i.e., weight loss, diet duration, sex, age, height, physical activity, health condition) for each post and successfully identified at least six or all seven domains in approximately 76% of the posts. Among 602 posts where weight loss was correctly identified, the LLM accurately identified at least five or all six remaining domains in 90% of cases.

Data are shown as median (25th; 75th percentile) for the continuous and as frequencies (*n* (%)) for the categorical variables. Histograms and P-P plots were used to evaluate whether data are normally distributed. Weight loss and diet duration data were Winsorized at the 1% and 99% percentile values to exclude extreme outliers, whilst age was truncated to 18–85 years old. Quantile regression and the test on equality of proportions were used, respectively, to calculate the 95% confidence intervals (CIs) for the differences in medians (for weight loss and age) and proportions (for diet duration, gender, workout and health conditions) between the available and complete cases. A linear regression model was used to examine the association between diet duration, age, gender, physical activity and health conditions with the weight loss following a ketogenic diet. Results are shown as regression coefficients (95% confidence interval, CI). All tests were two-sided, and the significance level was set to 0.05. Stata v.18 (StataCorp. 2023. Stata Statistical Software: Release 18. College Station, TX, USA: StataCorp LLC.) was used for the analysis.

## 3. Results

We have extracted 75,894 posts published from 29 July 2010 to 31 December 2023 in “r/keto” subreddit. On average, each post required 32 s to be processed, highlighting the computational demands of working with a large dataset using advanced natural language processing models.

Of them, 5312 (7%) posts were removed (*n* = 48 duplicate posts, *n* = 291 from the “DietTrackerBot”, *n* = 4954 from “deleted” accounts), leaving 70,601 (93%) posts for the analysis. Data for both weight loss and diet duration exist for approximately half of the cases (*n* = 35,050) (Figure 2).

Of the 35,050 valid posts with self-reported weight loss and diet duration data in the “r/keto” subreddit, the median (25th; 75th centile) is 9.5 kg (4.5; 20.4) and 53 days (20; 137) for weight loss and diet duration, respectively. Approximately 2/3 of posts (*n* = 27,213) in the “r/keto” subreddit refer to self-reported diet duration of up to 3 months, whilst only 5% reported adherence to a ketogenic diet for over a year. Age was self-reported in about 18% of the posts (*n* = 6498) with a median (IQR) of 30 (24; 38) years, whilst the LLM identified gender in 25% (*n* = 9084) of these posts with a slightly higher representation of males (52% vs. 48%). Lastly, about 1/3 (*n* = 11,994) of these posts self-reported an active lifestyle, and a similar proportion self-reported health conditions (*n* = 10,475).

Approximately 93% of the posts (*n* = 32,637) were excluded due to missing data in the additional covariates, thereby leaving 2413 (7%) posts with complete data for the analysis. In general, the distribution of weight loss, age and workout is similar between the excluded and the complete cases sample (Table 2). However, the proportion of males is higher in the complete cases sample (62.3% vs. 48.7%), whilst slightly more individuals reported any health condition (33.7%) compared to the available sample (29.6%). Additionally, diet duration in the complete cases sample is slightly longer (61 (21; 168)) compared to the excluded cases (49 (18; 122)), whilst a slightly higher percentage reported longer-term diet durations (>180 days) and fewer short-term diets (up to 30 days) (Table 2).

Within the complete cases sample (*n* = 2413), at least one health condition was mentioned in 813 posts. From these, we identified the 50 most frequently reported diseases and categorized them into 12 broader groups according to ICD-10 classification (Figure 3) [23]. “Endocrine, nutritional, and metabolic diseases” (ICD-10 Chapter 4), such as pre-diabetes, diabetes (type 1 and 2), insulin resistance, obesity, polycystic ovary syndrome, thyroid issues, and hypothyroidism, were the most self-reported conditions, followed by “Mental and behavioral disorders” (ICD-10 Chapter 5, i.e., ADHD, anxiety, binge eating, depression) and diseases of the “Circulatory system” (ICD-10 Chapter 9, i.e., hypertension, dyslipidemia) (Figure 3).

Weight loss significantly differed across diet duration categories (*p* = 0.0001) with longer durations associated with greater weight loss. Participants following a diet for more than 365 days experienced the highest median weight loss, while those dieting for ≤30 days lost the least. Weight loss also differed between genders (*p* = 6.3 × 10^−29^) with males losing more weight than females. Individuals with any health condition had significantly higher weight loss compared to those without (*p* = 4.5 × 10^−6^). However, lifestyle (sedentary vs. active) was not associated with significant differences in weight loss (*p* = 0.62) (Table 3).

Linear regression analysis revealed a significant association between weight loss and the duration of adherence to the ketogenic diet. Authors of the “r/keto” subreddit who followed the diet for over a year reported an average weight loss of 28 kg (95% CI: 25.5 to 30.9) compared to those with up to a month diet duration. Furthermore, male authors and those with reported health conditions had significantly greater weight loss compared to female authors and those without reported health conditions, respectively (Table 4).

## 4. Discussion

In this study, we explored the potential of utilizing large language models and the richness of the online platforms for dietary assessment. The results of this study confirm the well-documented effectiveness of the ketogenic diet in facilitating weight loss [24], particularly for individuals seeking rapid weight reduction or improved glycemic control [25], albeit the long-term health implications of the ketogenic diet remain controversial and merit periodic reassessment, particularly when adhering to the diet for longer periods [26].

Traditional research methods, such as recruiting participants, collecting measurements, and conducting surveys or interviews, are often resource-intensive, time-consuming, and limited by sample size and geographic scope. Researchers are increasingly turning to social media as a source of information investigating a wide range of health-related topics. This trend can be attributed to the public accessibility of these platforms, the relative ease of data acquisition (via platform-provided application programming interfaces), and the continuous contribution of geographically and demographically diverse user populations [27]. With the recent emergence of artificial intelligence [28] and the increasing computing power to process large amounts of data, state-of-the-art natural language processing techniques, with the implementation of LLMs, are now available to efficiently analyze large amounts of free-text data with minimal manual effort. For instance, in this study, we accessed approximately 75,000 posts discussing the impact of ketogenic diet on weight loss, which is a scale that would be nearly impossible to achieve through traditional methods in such a short timeframe. Even after narrowing the dataset to approximately 2500 posts for detailed analysis, the volume and diversity of data remain unparalleled compared to what could realistically be gathered from individuals. LLMs not only process this vast amount of unstructured data efficiently but also extract meaningful insights that would otherwise require significant effort and resources to obtain through conventional approaches. This highlights the potential of AI-driven tools to complement or even redefine traditional research methodologies in nutrition and health sciences.

While Reddit provides access to diverse user experiences, its discussions may be influenced by third-party interests, including the commercial promotion of diets, supplements, and ideological affiliations [29,30]. Given these potential biases, social media data should be interpreted with caution and complemented with traditional research approaches where possible. However, utilizing data from online platforms and forums is advantageous compared to traditional data collection methods, such as surveys or interviews. The anonymity of these platforms has the potential to reduce the impact of social desirability bias, encouraging more honest self-reporting of dietary habits, challenges and health outcomes [31,32].

Nonetheless, the representativeness of data collected from online communities must be carefully considered. Active participants in such forums may not perfectly reflect the broader general population, as they are often more motivated, tech-savvy, and driven by specific goals. The observed sex distribution, for instance, may reflect differences in engagement patterns within digital health communities, where men can be overrepresented in weight loss and fitness discussions [33,34]. Additionally, online diet discussions may attract individuals with specific health conditions or weight management goals, potentially introducing biases in self-reported experiences. While our findings align with trends observed in other digital health studies, the generalizability of AI-extracted cohorts from social media remains a key limitation. Differences in self-selection, motivations for dietary changes, and the overrepresentation of certain demographics may introduce selection bias. To enhance the robustness of future research, integrating AI-extracted data with traditional clinical and epidemiological samples could provide a more comprehensive understanding of diet-related behaviors across diverse populations. Moreover, as with any study relying on self-reported data, our findings are subject to recall and reporting biases, as well as self-selection effects inherent to online platforms. While our study aimed to analyze naturally occurring diet-related discussions, we recognize that online narratives may not fully reflect real-world dietary behaviors or physiological outcomes. Furthermore, while long-term adherence to the ketogenic diet remains debated, our study did not aim to validate its feasibility but rather to capture user-reported experiences. While these limitations may influence the magnitude of observed effects, the dataset’s composition aligns with established patterns in digital health communities, supporting its plausibility as a reflection of engaged users in online diet discussions. Given that highly motivated individuals often shape online health conversations, similar trends—albeit of smaller magnitude—may still be observed in the general population.

Variability in metabolic responses, initial weight status, and self-reporting inconsistencies may also explain unexpected trends in physical activity and weight loss. Furthermore, self-reported data inherently lack biochemical verification, making it impossible to confirm actual dietary adherence or what participants consumed. Although this limitation is inherent in any study dealing with human participants, our analysis focused on perceived experiences rather than objective dietary adherence. Prior research suggests that online diet discussions may amplify positive outcomes while underreporting adherence challenges and potential side effects, which is a trend observed in both social media-based and traditional self-reported dietary data [35]. While factors such as caloric intake, dietary quality, and metabolic adaptations may influence individual weight loss trajectories and adherence, studying these was beyond the scope of our study. Future research employing more structured designs could further investigate these aspects, potentially incorporating validation approaches such as cross-referencing self-reported weight changes with wearable health devices or external databases to mitigate biases.

As such, the primary aim of this study was not to estimate the effectiveness of the ketogenic diet in weight loss or to validate individual health claims but rather to leverage large-scale user-generated content to explore self-reported experiences. However, it is important to acknowledge that the substantial proportion of missing data precluded the use of advanced methods such as multiple imputation [36] and may have influenced the observed patterns, as users who provided complete information may differ systematically from those with incomplete reports. For example, the observed differences between excluded and complete cases highlight a potential selection bias, particularly regarding gender distribution in reported weight loss. While the LLM demonstrated robust classification performance, the shift in gender ratio suggests that missingness may not be entirely random. Future research should explore strategies to mitigate this bias, such as refining data extraction methodologies to account for missing covariates. While this limits the generalizability of the findings, it is reasonable to anticipate that similar effects, though potentially smaller in magnitude, may exhibit in the general population, supporting the broader relevance of these findings in health and nutrition research [37].

The performance of LLMs in extracting data is promising but not without challenges. LLMs can process large volumes of text and identify patterns or information domains efficiently, making them valuable tools for tasks such as dietary data extraction. However, errors and inaccuracies are inevitable. For instance, LLMs may misinterpret context, conflate overlapping information, or infer details that are implied but not explicitly stated. In our study, the model was not able to perform well in cases where a single post discussed multiple individuals or multiple changes over time and often assumed conditions, such as gender or health status, without explicit mentions. While the model demonstrated notable success in extracting comprehensive information, its lower accuracy in identifying key variables like weight loss and diet duration, both central to the study, highlights the need for refinement and validation to ensure reliability in extracting critical dietary information. The complexity of user-generated content, including vague language, fluctuations in reporting, or incomplete information, further compounds these challenges. Although the misclassification patterns we observed suggest that LLMs may struggle with contextual nuances, requiring further refinements in natural language processing techniques, we anticipate that testing larger versions of Llama 3 (e.g., Llama 3-70B) or alternative state-of-the-art LLMs trained on long-form text (e.g., GPT-4, Claude, or Mistral) could enhance contextual understanding and thereby reduce misclassification errors.

Validation remains critical to mitigate the impact of minor inaccuracies, which could lead to significant misinterpretations. In our study, 1% of the available posts were randomly selected and used for validation, offering sufficient power to estimate the LLM’s accuracy and generalize findings to the broader dataset under the assumption of a representative sample. However, in highly diverse datasets with substantial variability in style, length, or complexity, 800 posts may not fully capture all variations. Regardless, internal evaluation suggested that accuracy stabilizes after approximately 300 posts, therefore validating more posts would not significantly alter our accuracy estimates. This finding suggests that the misclassification patterns described above are not easily corrected by manual review, which was likely due to the inherent complexity of user-generated content. Ambiguous wording, implicit references, and inconsistent reporting of weight changes contribute to these challenges, as discussed in the previous paragraph. While validation efforts provided valuable insights, they did not significantly improve accuracy. Future studies could further mitigate these errors by fine-tuning LLMs on domain-specific datasets or incorporating hybrid AI–human validation frameworks to improve extraction reliability. Additional strategies, such as repeated sampling or bootstrap analyses, could enhance insights into the model’s accuracy, though these methods are time consuming and require significant human effort. Addressing these limitations will necessitate a combination of model improvement, rigorous validation, and careful consideration of the complexities inherent in analyzing unstructured text data.

The effectiveness of automated text analysis depends significantly on carefully designed prompts that align with the complexities of user-generated content, including vague language and incomplete information. The model’s output was heavily influenced by the way prompts were framed. Prompts lacking specific guidance on handling posts describing multiple individuals or fluctuating, inconsistent data were a key factor contributing to the model’s lower accuracy. This underscores the need for prompt designs that are both precise and context-aware to enhance the reliability of the LLM’s performance and mitigate errors in extracting critical information. Moreover, duplicate cases/comments could be a concern in social media analytics (SMA) as they can introduce bias. In our analysis, the dataset was restricted to a single subreddit (r/keto), minimizing the risk of widespread duplication across communities, whilst we only identified a small fraction of duplicates in the available data. Regardless, in future research when incorporating data from multiple subreddits, username matching and text similarity algorithms should be used to identify highly similar posts (i.e., posts with near-identical wording across different subreddits).

The processing time and hardware constraints highlight additional key limitations when working with LLMs and extensive datasets. The execution time of the code and the computational resources required for processing the dataset represent significant practical considerations for this study. On average, each post took approximately 32 s to process using the Llama-3-8B-Instruct model. This runtime includes preprocessing, model inference, and structured output extraction. These models are not only CPU-intensive but also highly memory-intensive, which necessitated the use of the 8-billion-parameter version of the Llama model. Although this smaller model provided satisfactory results, the availability of greater memory would allow the use of larger models with more parameters, potentially improving accuracy.

Future studies, which may include larger datasets or real-time dietary monitoring, could address these limitations by leveraging cloud-based platforms or high-performance computing (HPC) clusters equipped with faster CPUs and the processing power of GPUs, which are well suited for deep learning workloads. Such resources would enable the parallel processing of large datasets, reducing runtime and enhancing efficiency. Furthermore, the availability of larger memory in HPC setups would facilitate the use of more sophisticated models, further increasing the precision of the results. It is worth noting that there is significant effort from the scientific community to reduce the hardware requirements of LLMs while retaining comparable accuracy. Approaches such as model quantization, pruning, and distillation are being actively explored to create lightweight versions of these models. These advancements hold promise for making high-accuracy LLMs more accessible and practical for researchers with limited computational resources [38]. Further efficiency gains may be achieved by optimizing prompt engineering, reducing token redundancy, or fine tuning a larger but more efficient model such as Llama-3-13B. An alternative approach would be to employ a two-stage processing pipeline, where a lightweight model first filters relevant posts before applying a computationally intensive model for detailed analysis.

Ethical considerations are critical when using social media data and AI in research, particularly in sensitive areas like health and dietary behaviors. One key concern is informed consent and user privacy, as social media data are often treated as public, but their use raises questions about whether users have truly consented to their data being analyzed. Stirling et al. (2021) highlight that less than 25% of studies using SMA in nutrition research sought formal ethics approval, underlining the need for clear ethical guidelines [6]. In this study, we adhered to institutional ethical standards and GDPR regulations by processing the dataset to ensure anonymity, avoiding any personally identifiable information, and working only with publicly available posts. While the data are publicly accessible, we acknowledge the ethical responsibility to respect user privacy and ensure that the data are used in a manner consistent with both legal and ethical standards. Transparency in data collection and analysis is also vital, especially given the “black box” nature of many AI systems, which makes it difficult to trace how insights are derived or how individual data are used. This important gap in formal ethics reviews for SMA studies raises concerns about shared responsibility across researchers, developers, and decision makers when combined with AI. To address these challenges, and integrate SMA and AI responsibly, researchers must develop comprehensive ethical guidelines, enhance transparency in methods, mitigate bias through diverse datasets, prioritize formal ethical review processes, and evaluate the long-term impacts of their findings on individuals and communities. To mitigate biases and enhance interpretability in future implementations, we propose integrating human-in-the-loop validation, where domain experts review a subset of AI-extracted data to refine the model’s outputs iteratively. Additionally, employing ensemble modeling techniques, where multiple AI models cross-validate extracted variables, could further enhance robustness and reduce model-specific biases. As part of ongoing improvements, we are also considering fine tuning Llama-3 on domain-specific dietary datasets, which may improve contextual understanding and extraction accuracy.

In the realm of dietary assessment, the integration of AI and LLMs presents both exciting opportunities and significant challenges, particularly concerning model interpretability and explainability. Given the black-box nature of many LLMs, researchers must prioritize the development of tools and frameworks that elucidate how these models derive their conclusions. For instance, techniques such as attention visualization can help demystify the decision-making process of LLMs, allowing researchers to verify that the extracted dietary patterns and insights are both accurate and meaningful. In our study, for example, we considered all posts under the “r/keto” subreddit assuming that all were relevant to the ketogenic diet. Such transparency is crucial for building trust in AI-generated outputs, especially when these insights are used to inform public health policies or individual dietary recommendations. Furthermore, LLMs should be viewed as complementary to traditional dietary assessment methods, such as food frequency questionnaires or 24 h dietary recalls, to inform recommended dietary practices and food policies [39]. By integrating LLM-driven insights with survey data, researchers can validate findings and enrich their analyses with nuanced perspectives that might otherwise be overlooked. As LLMs evolve, they could facilitate breakthroughs in personalized nutrition and public health surveillance, enabling more scalable and accessible research tools. However, challenges remain, particularly regarding the potential for disparities in access to these advanced technologies. Barriers due to socioeconomic factors or technological limitations underscore the need for equitable solutions in the deployment of AI in health research.

## 5. Conclusions

Large language models (LLMs) are revolutionizing how we study diet by allowing us to analyze vast amounts of information. They can uncover hidden patterns in dietary data, which can lead to better public health strategies and personalized nutrition plans. However, it is crucial to carefully check the accuracy of the information LLMs find. We need to compare their results to more traditional methods of assessing diet, such as carefully tracking what people eat for a day or over a longer period. This helps ensure the information we obtain from LLMs is reliable and useful. Moreover, we need to be aware that information from online sources might not accurately reflect the dietary habits of everyone. Therefore, we need to explore diverse sources and combine LLM-based findings with data from more representative groups of people to develop more effective and equitable dietary interventions that address the diverse needs of different communities. Future research should prioritize hybrid approaches that integrate AI-driven analyses with established epidemiological and clinical methods to enhance both validity and applicability in real-world settings.

## Figures and Tables

**Figure 1 nutrients-17-00818-f001:**
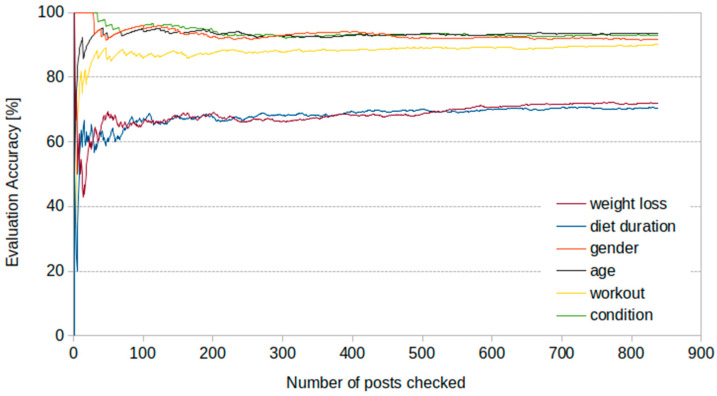
Classification performance of the LLM for all parameters.

**Figure 2 nutrients-17-00818-f002:**
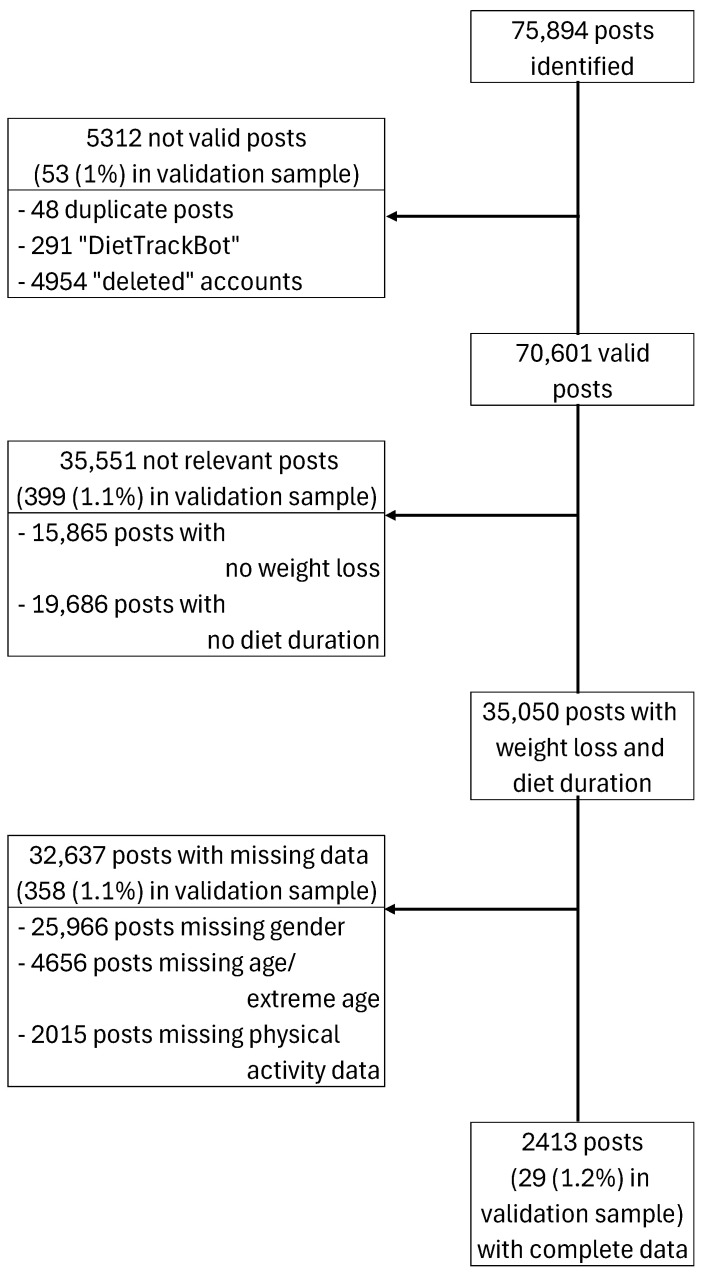
Flowchart of posts included in the study.

**Figure 3 nutrients-17-00818-f003:**
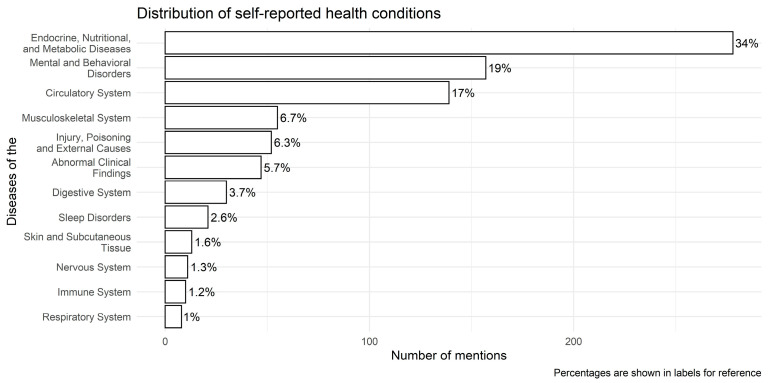
Distribution of self-reported health conditions in the complete cases sample (*n* = 2413).

**Table 1 nutrients-17-00818-t001:** The prompt used to guide the model’s output.

I need to extract numerical information from the following Reddit post. For each post, please provide:
Weight Lost: The amount of weight lost. If the post explicitly mentions the weight lost, provide that value with its relevant units (e.g., kg, lbs, pounds, etc.). If only initial and current weights are mentioned, calculate the difference and provide the result. Do not include any additional explanations.
Duration: The period over which the weight loss occurred. Do not include any additional explanations.
Gender: Indicate if the user is a man or woman. Do not include any additional explanations.
Age: Provide the user’s age in years. Do not include any additional explanations.
Height: Provide the user’s height. Do not include any additional explanations.
Works out: Indicate whether the user works out or not. Do not include any additional explanations.
Conditions: Any health conditions or issues the user mentions (e.g., injuries, illnesses). Do not include any additional explanations.
If the information is not available respond with “not applicable.” Do not include any additional explanations.

**Table 2 nutrients-17-00818-t002:** Descriptive characteristics of “r/keto” subreddit users (*n* = 35,050).

Domain	Median (p25; p75) ^1^		Mean Difference (Complete-Missing)
Self-Reported	Missing Covariate Data (*n* = 32,637)	Complete Cases (*n* = 2413)	(95% Confidence Interval, CI)
Weight loss, kg	9.5 (4.5; 20.0)	10.9 (4.5; 22.7)	1.4 (0.9; 1.8)
Diet duration, *n* (%)			
Up to 30 days	13,717 (42.0)	875 (36.3)	-
31 days to 90 days	6695 (20.5)	505 (20.9)	0.4% (−1.2%; 2.1%)
91 days to 180 days	5339 (16.4)	405 (16.8)	0.4% (−1.1%; 2.0%)
181 days to 365 days	5243 (16.1)	441 (18.3)	2.2% (0.6%; 3.8%)
>365 days	1643 (5.0)	187 (7.8)	2.7% (1.6%; 3.8%)
Gender, *n* (%)			
Female	3421 (51.3)	909 (37.7)	-
Male	3250 (48.7)	1504 (62.3)	13.6% (11.6%; 15.6%)
*Missing/not reported*	*25,966*	-	
Age, years	30 (25; 39)	29 (24; 36)	−1.0 (−1.5; −0.5)
*Missing/extreme age*	*28,552*	-	
Workout, *n* (%)	10,039 (80.9)	1955 (81.0)	0.1% (−1.5%; 1.7%)
*Missing/not reported*	*20,229*	-	
Health condition, *n* (%)	9662 (29.6)	813 (33.7)	4.1% (2.1%; 6.0%)

^1^ Data are shown as median (25th; 75th percentile), unless otherwise indicated. Italics were used to indicate missing data excluded from the analysis. Quantile regression was used to calculate the CIs for the differences in medians between the two groups for weight loss and age. The test on the equality of proportions was used to calculate the CIs for the differences in proportions for diet duration, gender, workout and health condition.

**Table 3 nutrients-17-00818-t003:** Descriptive statistics (median (1st percentile; 3rd percentile)) for weight loss across various sociodemographic characteristics of the complete cases sample (*n* = 2413).

	*n*	Weight Loss, kg	*p*-Value ^1^
Diet duration			0.0001
up to 30 days	875	4.5 (2.7; 8.6)	
31 to 90 days	505	9.1 (4.5; 13.6)	
91 to 180 days	405	15.6 (10.9; 22.7)	
181 to 365 days	441	25.8 (16.8; 37.2)	
>365 days	187	34.0 (20.4; 49.9)	
Gender			6.3 × 10^−29^
Male	1504	13.1 (6.3; 25.5)	
Female	909	7.3 (3.2; 16.8)	
Lifestyle			0.62
Sedentary	458	11.7 (5.0; 21.3)	
Active	1955	10.5 (4.5; 22.7)	
Health condition			4.5 × 10^−6^
None	1600	10 (4.5; 21.9)	
Any	813	12.8 (5.4; 25.4)	

^1^ *p*-values were calculated using the Kruskal–Wallis rank test for diet duration comparisons and the Wilcoxon rank-sum test for all other comparisons.

**Table 4 nutrients-17-00818-t004:** Results of linear regression model examining the association of sociodemographic and lifestyle covariates with weight loss in the complete cases sample (*n* = 2413).

	b (95% CI) ^1^	*p*-Value
Self-Reported Co-Variates		
Diet duration (vs. up to 30 days)		
31 to 90 days	3.2 (1.4; 5.1)	0.0007
91 to 180 days	8.8 (6.8; 10.8)	1.96 × 10^−17^
181 to 365 days	20.4 (18.4; 22.4)	1.55 × 10^−84^
>365 days	28.2 (25.5; 30.9)	1.66 × 10^−85^
Male vs. female	5.2 (3.8; 6.6)	1.00 × 10^−12^
Age, per year older	−0.05 (−0.12; 0.02)	0.16
Active vs. sedentary lifestyle	−0.11 (−1.9; 1.6)	0.89
Health condition vs. none reported	2.2 (0.7; 3.6)	0.004

^1^ Results are shown as regression coefficients, b, alongside 95% confidence intervals.

## Data Availability

All data used in this study consist of anonymized Reddit posts from the “r/keto” subreddit and are publicly available on Academic Torrents repository. Researchers can access the dataset from https://academictorrents.com/details/56aa49f9653ba545f48df2e33679f014d2829c10, (accessed on 17 September 2024) following the terms and conditions set by its original uploader. No additional data were generated in this study.

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
