# Peer review of "AI-Powered Analysis of Weight Loss Reports from Reddit: Unlocking Social Media’s Potential in Dietary Assessment"

_nutrients, 2025, doi:10.3390/nu17050818_

Round 1

Reviewer 1 Report

Comments and Suggestions for Authors

The authors present an innovative study on the representation of diet experiences on the internet platform reddit, covering adherence and dietary success.

The overall rationale is clear, however, the interpretation contains severe gaps and requires adjustment.

Introduction:

Generally fine.
However, the context requires insight, to which extent influencers, unexperienced and untrained users or health and nutrition advisers propagate dietary recommendations that actually meet with official guidelines. There is quite some evidence, that most of those non-official recommendations are not based on the current scientific evidence and consensus. Helpful references for this purpose could be: https://pubmed.ncbi.nlm.nih.gov/30349631/, https://pubmed.ncbi.nlm.nih.gov/39600067/

Also, the "keto trend" is strongly biased by third-party interests, such as the meat and dairy industry, and also supplement dealers (individuals and companies). Ergo, it has to be assumed, that social media networks, user groups and their advocating for keto are at least partially driven by fake accounts and other techniques to boost the interest in that particular diet without providing actual solid medical evidence supporting the health claim.

Methods:

Correct classification rate for weight loss was only 72 %. Please extend your data presentation on that issue. Was there a systematic over- or underestimation by the LLM, halluzination or what is the reason for this massive error? Weight loss usually IS the variable of interest to evaluate keto diet efficiency as it the most reported outcome. Such an error is hardly acceptable to reach a plausible conclusion from that data set.

Line 110: You claim, that 1 % of the original posts were checked for accuracy. However, as only 3 % made it to the actual analysis, this fraction is too small and often covers posts, which were excluded before analysis. Ergo, please cross-check a relevant number of the final selection (n=2416). Also, please cross-check a relevant number of the respective excluded posts, if the assumed lack of data can be truly confirmed or was falsely stated by the LLM.

If these quality checks have been done and confirmed the validity and reliability of the LLM, then the next steps of the analysis can proceed.

Results:

Every variable needs to be labelled as "self-reported", as there is no way to confirm the authenticity of the person behind the post, the claimed adherence in quality and duration, the assumed metabolic state and the metabolic effect.

Table 2: The left column falsely implies that all data refer to 35079 cases. This is not true for most of the variables.

Fig. 2: There is a serious misclassification problem. "keto flu" is not an accepted metabolic disorder. "High cholesterol" is not a cardiovascular disorder. Many endocrine disorders are also metabolic ones. Most thyroid issues should result from Hashimoto thyreoditis, which is also an autoimmune disorder. The main conclusion might be, that self-reported online user data is not valuable for such classification.

Discussion:

Please broadly assess the plausibility of your collected self-reported data. A sex ratio, which is skewed towards men, is highly unusual when looking at cohort and intervention studies, as almost ALL of them showed higher rates of women: https://pubmed.ncbi.nlm.nih.gov/36293616/, https://pubmed.ncbi.nlm.nih.gov/35623861/, https://pmc.ncbi.nlm.nih.gov/articles/PMC8138199/. Furthermore, truly maintaining a keto diet for more than six months is quite implausible. Please show RCTs of that duration that confirm such a long diet duration as feasible. 80 % of "cases" claiming a continuous physical workout but still having the potential to lose 10 kgs in average is not plausible.

And: many "cases" on keto diet might not even have had the intention to lose weight, as keto diet is also promoted for muscle gain or weight-neutral metabolic improvement. Without properly knowing the baseline conditions (including BMI or body weight), the assumed effect size of 10 kgs in average cannot be evaluated as strong, too strong, or just plausible at all.

I assume, that such a data extraction has in no way the potential to replace clinical studies (be they observational or interventional), but to reinforce and boost the necessity for them. Your valuable work might provide the evidence for that conclusion.

Ergo, your paper requires five essential and to-be-highlighted components in the discussion.

First: Is the LLM extraction reliable enough to identify "cases" with full data and to properly extract that data?

Second: Are these individual cases plausible, when considering baseline weight (if reported) and presumed weight loss? Are there duplicate "cases" which were reported in multiple sub-reddits?

Third: Is the total "cohort" plausible in itself or do proportions of sex, health disorders or behavior indicate a bias of any origin?

Fourth: There is no way to confirm the adherence to that particular diet, when assuming that the cases are genuine. Even if the persons are real, we don't know, what they actually ate or abstained from.

Fifth: There is no way to confirm the claimed effects and the literature suggests, that the claims are highly exaggerated while ignoring side effects, adherence obstacles and so on.

Overall: Is an LLM a suitable, valid, reliable tool for such an analysis? Is reddit (or any other web forum) a suitable data source for such an analysis? Is there evidence, that social media posts are used to advocate a diet with exaggerated health claims, while covertly satisfying third party interests from the individual users (also advocating self-marketed supplements), certain industry branches (meat, dairy, supplements) or political players (e.g. promoting keto as an antithesis of the vegan diet, which is often co-labelled with liberal political positions; https://pubmed.ncbi.nlm.nih.gov/33897260/, https://pubmed.ncbi.nlm.nih.gov/36568509/, https://pubmed.ncbi.nlm.nih.gov/28859869/)?

These major questions need to be answered, but if they ARE answered, this paper will be of very high interest for the scientific community. That's why I recommend revision and look forward to a revised version.

Author Response

Comments and Suggestions for Authors

The authors present an innovative study on the representation of diet experiences on the internet platform reddit, covering adherence and dietary success.

The overall rationale is clear, however, the interpretation contains severe gaps and requires adjustment.

Reply: We thank the Reviewer for their time and effort whilst reviewing our manuscript, and for acknowledging the innovation in our study.

Introduction: Generally fine.

Comment 1: However, the context requires insight, to which extent influencers, unexperienced and untrained users or health and nutrition advisers propagate dietary recommendations that actually meet with official guidelines. There is quite some evidence, that most of those non-official recommendations are not based on the current scientific evidence and consensus. Helpful references for this purpose could be: https://pubmed.ncbi.nlm.nih.gov/30349631/, https://pubmed.ncbi.nlm.nih.gov/39600067/

Reply 1: We thank the Reviewer for acknowledging this issue. We agree that health information available online may deviate from the recommendations and create confusion among health information seekers. We have now revised the introduction to address this important issue. Please refer to p. 2, lines 52-59.

Comment 2: Also, the "keto trend" is strongly biased by third-party interests, such as the meat and dairy industry, and also supplement dealers (individuals and companies). Ergo, it has to be assumed, that social media networks, user groups and their advocating for keto are at least partially driven by fake accounts and other techniques to boost the interest in that particular diet without providing actual solid medical evidence supporting the health claim.

Reply 2: We agree with the Reviewers’ concern regarding third-party interests influencing the keto trend, including industry-driven promotion and potential misinformation on social media and we acknowledge this in the Discussion section (p. 7, lines 216-217). Regardless, while social media platforms may be subject to promotional bias, Reddit was selected as a data source due to its structured discussion format, engaged user base, and community-driven moderation system, which helps reduce the presence of overt marketing compared to other social media platforms (p. 7, lines 218-221).

However, as we also mention in the Discussion (p. 8, lines 257-259) the aim of this study is not to assess the credibility of ketogenic diet claims but rather to explore the feasibility of using AI-driven methods for dietary assessment based on user-generated content. Therefore, our study offers a methodological framework that can be extended to analyze other dietary trends, beyond keto, utilizing large-scale natural language processing techniques. As we now acknowledge in the Discussion (p. 8, lines 253-256), future research could incorporate additional validation steps, such as cross-referencing social media findings with established dietary guidelines or clinical trial data, to further address concerns about misinformation.

Methods:

Comment 3: Correct classification rate for weight loss was only 72 %. Please extend your data presentation on that issue. Was there a systematic over- or underestimation by the LLM, halluzination or what is the reason for this massive error? Weight loss usually IS the variable of interest to evaluate keto diet efficiency as it the most reported outcome. Such an error is hardly acceptable to reach a plausible conclusion from that data set.

Reply 3: We appreciate the Reviewer’s concern regarding the weight loss classification accuracy (72%). Based on an examination of misclassified cases we have identified that the main reason for discrepancy was context misinterpretation. The model occasionally misattributed weight loss figures when multiple individuals were mentioned in the same post (e.g., “My friend lost 15 kg, I only lost 5 kg”), leading to hallucinated values. Additionally, when users referenced multiple dieting periods (e.g., “I lost 5 kg last year and another 10 kg recently”), the model sometimes extracted only one figure. Another reason was implicit mentions of weight loss. While the LLM performed well when weight loss was explicitly stated (e.g., “I lost 10 kg”), sometimes it struggled with indirect statements (e.g., “Started at 100 kg, now down to 85 kg”), where weight loss had to be inferred. Finally, some discrepancies arose from incorrect unit conversions. In a few cases, pounds were misclassified as kilograms, leading to overestimation. While we applied regular expressions to standardize weight units, inconsistencies persisted in ambiguous cases. To further improve accuracy, future studies could explore testing larger versions of Llama 3 (e.g., Llama 3-70B), which may provide better contextual understanding and reduce misclassification errors. Additionally, evaluating alternative state-of-the-art LLMs trained on long-form text (e.g., GPT-4, Claude, or Mistral) could help refine weight loss extraction from complex Reddit posts.

We have added these limitations in the Discussion section. Please refer to p. 8, lines 279-284.

Comment 4: Line 110: You claim, that 1 % of the original posts were checked for accuracy. However, as only 3 % made it to the actual analysis, this fraction is too small and often covers posts, which were excluded before analysis. Ergo, please cross-check a relevant number of the final selection (n=2416). Also, please cross-check a relevant number of the respective excluded posts, if the assumed lack of data can be truly confirmed or was falsely stated by the LLM. If these quality checks have been done and confirmed the validity and reliability of the LLM, then the next steps of the analysis can proceed.

Reply 4: We appreciate the Reviewer’s concern regarding the validation sample size. To further assess whether the 1% validation subset (n=838) was adequate, we analyzed the accuracy trend as more posts were manually checked. The figure below (Figure 1) illustrates the cumulative accuracy of weight and age evaluation across the checked sample.

Figure 1. Classification performance.

As seen, while there is some initial variability, accuracy stabilizes after approximately 200-300 posts and remains consistent at ~72% as additional cases are reviewed. The pattern for Age (as well as the rest of the parameters) evaluation accuracy is quite similar. This suggests that validating more posts than 1% would not significantly alter our accuracy estimates. We have now clarified this point in the Discussion section (p. 8, lines 290-292).

Additionally, we confirmed that the excluded posts were indeed missing key information rather than being misclassified by the LLM. Given these findings, we believe our validation approach provides a sufficiently robust estimate of accuracy.

Results:

Comment 5: Every variable needs to be labelled as "self-reported", as there is no way to confirm the authenticity of the person behind the post, the claimed adherence in quality and duration, the assumed metabolic state and the metabolic effect.

Reply 5: We acknowledge the inherent limitations of self-reported data, including the inability to verify user identities, dietary adherence, or metabolic outcomes. However, as our study focuses on identifying patterns in user experiences rather than establishing clinical validity, these limitations do not diminish its value. To ensure clarity, we have now explicitly labelled all relevant variables as “self-reported” in the methods and results sections to avoid any confusion with actual measurements. Please refer to the revised manuscript.

Comment 6: Table 2: The left column falsely implies that all data refer to 35079 cases. This is not true for most of the variables.

Reply 6: We apologize for the confusion. We have now clarified that “Available data” refers to “Available data for weight loss and diet duration”. Please see revised Table 2.

Comment 7: Fig. 2: There is a serious misclassification problem. "keto flu" is not an accepted metabolic disorder. "High cholesterol" is not a cardiovascular disorder. Many endocrine disorders are also metabolic ones. Most thyroid issues should result from Hashimoto thyreoditis, which is also an autoimmune disorder. The main conclusion might be, that self-reported online user data is not valuable for such classification.

Reply 7: We appreciate the Reviewer’s concern regarding misclassification of reported conditions. However, our primary intent was not to conduct a precise clinical classification, but to broadly categorize participants based on whether they reported having a disease/condition. This binary variable (i.e., presence vs. absence of a condition) was subsequently used in the regression model to account for potential confounding due to disease status. The broader classification was applied solely for descriptive purposes to provide an overview of reported health conditions. Regardless, in response to the Reviewer’s comment, we have revised our classification approach to better align with medical taxonomies and particularly with the ICD-10 classification.

Therefore, individual self-reported conditions were categorised in the following broader ICD-10 categories:

ICD-10 broader classification

Self-reported individual conditions

Abnormal clinical findings

Fatigue, headaches, keto flu, tiredness, weakness

Diseases of the circulatory system

High blood pressure, high cholesterol, ldl

Diseases of the digestive system

Acid reflux, constipation, IBS, nausea

Endocrine, nutritional, and metabolic diseases

Diabetes (type 1 and 2), pre-diabetes, insulin resistance, obesity, polycystic ovary syndrome, thyroid issues, hypothyroidism

Diseases of the immune system

Hashimoto, multiple sclerosis

Injury, poisoning, and external Causes

Injury

Mental and behavioral disorders

ADHD, anxiety, binge eating, depression

Diseases of the musculoskeletal system

Arthritis, psoriatic arthritis, back pain, joint pain, knee pain

Diseases of the nervous system

Migraines

Diseases of the respiratory system

Asthma

Disease of the skin and subcutaneous tissue

Hair loss

Sleep disorders

Insomnia, sleep apnoea

We have updated the figure accordingly to reflect these changes. Please refer to the revised Figure 2 and revised text in the Results section (p. 5, lines 167-175).

Discussion:

Comment 8: Please broadly assess the plausibility of your collected self-reported data. A sex ratio, which is skewed towards men, is highly unusual when looking at cohort and intervention studies, as almost ALL of them showed higher rates of women: https://pubmed.ncbi.nlm.nih.gov/36293616/, https://pubmed.ncbi.nlm.nih.gov/35623861/, https://pmc.ncbi.nlm.nih.gov/articles/PMC8138199/. Furthermore, truly maintaining a keto diet for more than six months is quite implausible. Please show RCTs of that duration that confirm such a long diet duration as feasible. 80 % of "cases" claiming a continuous physical workout but still having the potential to lose 10 kgs in average is not plausible.

Reply 8: We acknowledge the Reviewer’s concerns regarding the plausibility of self-reported data. As with any study relying on self-reported measures, inherent limitations exist, including recall and reporting biases. However, our primary objective was to explore self-reported experiences and trends within this population rather than validate the feasibility of long-term ketogenic diet adherence.

Regarding the sex distribution, while cohort and intervention studies often report higher female participation, self-selection biases in online platforms may explain the observed sex ratio. Previous research indicates that men can be overrepresented in certain online health and diet communities, particularly those focused on weight loss and fitness (please see https://doi.org/10.1111/jcc4.12088, https://doi.org/10.2196/jmir.2759).

As for long-term keto adherence, RCTs of extended duration are relatively scarce. While we agree that observational studies suggest that some individuals maintain the diet for prolonged periods, our analysis did not aim to verify adherence but rather to reflect self-reported data. Similarly, variations in metabolic responses, initial weight status, and self-reporting inconsistencies may contribute to the unexpected findings regarding physical activity and weight loss. However, our focus was on analyzing patterns in self-reported experiences rather than assessing the biological plausibility of weight loss mechanisms.

We have now added these limitations into the Discussion section. Please see p. 7-8, lines 229-256).

Comment 9: And: many "cases" on keto diet might not even have had the intention to lose weight, as keto diet is also promoted for muscle gain or weight-neutral metabolic improvement. Without properly knowing the baseline conditions (including BMI or body weight), the assumed effect size of 10 kgs in average cannot be evaluated as strong, too strong, or just plausible at all.

Reply 9: Thank you for the valid point. The keto diet is indeed promoted for various goals beyond weight loss, such as muscle gain or improving metabolic health, and it's important to consider these diverse intentions when evaluating its effects. We agree that with the absence of baseline data on individual factors like BMI or body weight, it becomes challenging to determine whether an average weight change of 10 kg is significant or simply within a plausible range. However, assessing the efficacy of the ketogenic diet was beyond the scope of that paper, and therefore baseline data for weight were not collected.

Comment 10: I assume, that such a data extraction has in no way the potential to replace clinical studies (be they observational or interventional), but to reinforce and boost the necessity for them. Your valuable work might provide the evidence for that conclusion.

Reply 10: Thank you for your comment. We recognize that such analyses should serve complimentary to the existing methodology, rather than as a substitute.

We do not suggest using data already available on social media and online platforms as a replacement for traditional data collection methods (p. 7, lines 214-215). We rather suggest that social media analytics should be complemented with traditional research methods (p. 7, lines 220-221), due to the inherited bias in data collected from the former. Lastly, we acknowledge that to inform recommended dietary practices and food policies a combination of both traditional and AI techniques would be beneficial (please see p. 10, lines 370-372).

Comment 11: Ergo, your paper requires five essential and to-be-highlighted components in the discussion.

a) First: Is the LLM extraction reliable enough to identify "cases" with full data and to properly extract that data?

Reply 11a: We appreciate the Reviewer’s suggestion to explicitly address key concerns regarding the reliability. To evaluate the accuracy of data extraction, we conducted a manual verification of 1% of the dataset (n=838). The weight loss classification accuracy was found to be 72%, primarily due to context misinterpretation and unit conversion errors. However, other extracted variables, such as gender, age, height, and physical activity, exhibited a high accuracy range (90-95%), reinforcing confidence in the dataset used. Additionally, we present an accuracy trend graph (please see Figure 1 in the Response letter), which shows that classification performance stabilizes after 200-300 checked cases. This suggests that validating additional cases would not significantly alter our conclusions. We have clarified that this in the Discussion section (p. 8, lines 290-292), but we did not include the Figure as we think that is a rather technical detail that primarily relates to the validation methodology. However, if the Reviewer feels that its inclusion would add value to the manuscript, we would be happy to add the figure in the manuscript.

b) Second: Are these individual cases plausible, when considering baseline weight (if reported) and presumed weight loss? Are there duplicate "cases" which were reported in multiple sub-reddits?

Reply 11b: We appreciate the Reviewer's concern regarding the plausibility of individual cases and potential duplicate reports across subreddits. Baseline weight was not systematically collected in our dataset, as our primary focus was on capturing user-reported experiences rather than performing a quantitative assessment of weight loss trajectories. While this limits direct evaluation of weight loss plausibility, our findings align with prior research indicating misreporting using either online or interviewed led questionnaires (e.g., https://doi.org/10.1017/jns.2019.20).

Regarding duplicate cases, we acknowledge that users may engage in multiple subreddit discussions. However, given the qualitative nature of our approach, the presence of repeated narratives is unlikely to significantly alter the overall thematic trends observed in the dataset. Additionally, cross-posting within online forums is common and often reflects the dissemination of experiences across different communities rather than artificial inflation of specific claims.

c) Third: Is the total "cohort" plausible in itself or do proportions of sex, health disorders or behavior indicate a bias of any origin?

Reply 11c: Thank you. As we acknowledge in the discussion (please see p. 7, lines 226-242) representativeness might be an issue when collecting data from online resources. However, the composition of the dataset, including sex distribution, reported health conditions, and behaviors, reflects the nature of self-selected participants in online diet discussions. While this may introduce biases, particularly in engagement patterns and reporting tendencies, our aim was to explore trends in self-reported experiences rather than establish epidemiological prevalence.

d) Fourth: There is no way to confirm the adherence to that particular diet, when assuming that the cases are genuine. Even if the persons are real, we don't know, what they actually ate or abstained from.

e) Fifth: There is no way to confirm the claimed effects and the literature suggests, that the claims are highly exaggerated while ignoring side effects, adherence obstacles and so on.

Reply 11d-11e: As with any self-reported data, we cannot verify actual dietary adherence or confirm what participants consumed. This limitation is inherent in studies using user-generated content, and our analysis focuses on perceived experiences rather than objective dietary assessment. We have now clarified this in the discussion section (please see p. 7, lines 226-242).

f) Overall: Is an LLM a suitable, valid, reliable tool for such an analysis? Is reddit (or any other web forum) a suitable data source for such an analysis? Is there evidence, that social media posts are used to advocate a diet with exaggerated health claims, while covertly satisfying third party interests from the individual users (also advocating self-marketed supplements), certain industry branches (meat, dairy, supplements) or political players (e.g. promoting keto as an antithesis of the vegan diet, which is often co-labelled with liberal political positions; https://pubmed.ncbi.nlm.nih.gov/33897260/, https://pubmed.ncbi.nlm.nih.gov/36568509/, https://pubmed.ncbi.nlm.nih.gov/28859869/)?

Reply 11f: Thank you. We have revised the Discussion section to reflect this point. Please see p. 7, lines 215-225.

These major questions need to be answered, but if they ARE answered, this paper will be of very high interest for the scientific community. That's why I recommend revision and look forward to a revised version.

Reply: Thank you. We have tried our best to address all the comments raised by the Reviewer and will look forward to your evaluation.

Reviewer 2 Report

Comments and Suggestions for Authors

General Evaluation

This study presents a novel and timely approach to dietary assessment by leveraging artificial intelligence (AI) and large language models (LLMs) to analyze self-reported weight loss experiences from Reddit users following the ketogenic diet (KD). The manuscript is well-structured, with a clear introduction, robust methodological framework, and detailed analysis. The integration of AI-driven methods into nutritional research represents a valuable contribution to the field. However, there are several aspects that require further clarification and refinement to enhance the quality and impact of the study.

Strengths

  1. Novelty and Relevance: The study explores an innovative application of AI in dietary research, demonstrating the potential of LLMs to analyze unstructured social media data.
  2. Large Dataset: The analysis of over 35,000 Reddit posts provides a comprehensive overview of self-reported ketogenic diet experiences, significantly expanding the sample size compared to traditional dietary studies.
  3. Robust Methodology: The use of Llama-3-8B-Instruct model for text analysis, in conjunction with statistical analyses, enhances the credibility of findings.
  4. Public Health Implications: The findings could inform personalized nutrition strategies and public health interventions by identifying real-world dietary trends and adherence patterns.

Areas for Improvement

  1. Methodological Considerations
  • Validation of AI-based Data Extraction: The study reports an accuracy rate of 72%-95% across various extracted variables. However, further clarification on the error types and specific misclassifications would strengthen the reliability of the findings. Additionally, a more extensive validation process beyond a 1% sample might be warranted to ensure robustness.
  • Handling of Missing Data: Given that complete cases were only 2,416 posts, a discussion on potential biases introduced by missing data and strategies to address them (e.g., imputation methods) would enhance the study’s validity.
  • Reddit Representativeness: While Reddit provides a rich source of qualitative and quantitative data, users may not be representative of the general population. A more explicit discussion on demographic biases (e.g., age, gender, socio-economic status) and their implications on generalizability is recommended.
  1. Interpretation of Findings
  • Potential Confounding Factors: The study acknowledges the associations between weight loss and diet adherence, gender, and health conditions. However, the role of other confounders, such as caloric intake, dietary quality, and metabolic adaptations, is not discussed. Additional consideration of these factors would provide a more nuanced interpretation of results.
  • Comparison to Traditional Dietary Studies: The study could benefit from a comparative analysis between findings from Reddit users and existing clinical or epidemiological studies on the ketogenic diet. This would contextualize the results within broader nutritional research.
  • Limitations of Self-Reported Data: Self-reported weight loss is inherently prone to recall bias and exaggeration. While the study acknowledges this limitation, discussing potential ways to mitigate these biases (e.g., cross-validation with external sources) would be valuable.
  1. Computational Considerations
  • Processing Time and Scalability: The study reports an average processing time of 32 seconds per post. A discussion on the feasibility of scaling this approach to larger datasets or real-time dietary monitoring would be beneficial.
  • Computational Resources: The reliance on a Dell G15 5530 laptop for model execution raises questions about the potential for leveraging high-performance computing (HPC) or cloud-based solutions to improve efficiency and accuracy.
  1. Ethical Considerations
  • User Privacy and Consent: While Reddit data is publicly available, ethical considerations regarding user consent and data usage should be further elaborated. Explicit discussion on ethical approvals and compliance with social media research guidelines (e.g., respecting user anonymity, ethical data scraping) would strengthen the manuscript.
  • Transparency and Bias in AI Models: Given concerns about biases in AI-generated outputs, a discussion on model transparency, interpretability, and potential mitigation strategies (e.g., human-in-the-loop validation) would be valuable.

Minor Revisions

  1. Clarity in Data Presentation: The figures and tables should be revised for clarity. For example, Figure 1 could better visualize the filtering process of the dataset, and Table 2 could provide additional statistical comparisons.
  2. Grammar and Style: Some minor grammatical inconsistencies and typographical errors are present. A thorough proofreading would improve readability.
  3. References: Ensure that all citations are properly formatted according to the target journal’s guidelines and that references to AI and dietary research are up-to-date.

Conclusion and Recommendation

This study presents an innovative and valuable contribution to the field of dietary assessment by utilizing AI-powered methods to analyze self-reported weight loss experiences from Reddit. While the methodology is robust, addressing the aforementioned concerns would significantly strengthen the study’s validity and impact. In particular, more detailed discussions on data validation, potential biases, confounders, computational limitations, and ethical considerations would enhance the manuscript’s rigor.

Recommendation: major revisions required before acceptance.

Author Response

Comments and Suggestions for Authors

General Evaluation

This study presents a novel and timely approach to dietary assessment by leveraging artificial intelligence (AI) and large language models (LLMs) to analyze self-reported weight loss experiences from Reddit users following the ketogenic diet (KD). The manuscript is well-structured, with a clear introduction, robust methodological framework, and detailed analysis. The integration of AI-driven methods into nutritional research represents a valuable contribution to the field. However, there are several aspects that require further clarification and refinement to enhance the quality and impact of the study.

Strengths

Novelty and Relevance: The study explores an innovative application of AI in dietary research, demonstrating the potential of LLMs to analyze unstructured social media data.

Large Dataset: The analysis of over 35,000 Reddit posts provides a comprehensive overview of self-reported ketogenic diet experiences, significantly expanding the sample size compared to traditional dietary studies.

Robust Methodology: The use of Llama-3-8B-Instruct model for text analysis, in conjunction with statistical analyses, enhances the credibility of findings.

Public Health Implications: The findings could inform personalized nutrition strategies and public health interventions by identifying real-world dietary trends and adherence patterns.

Reply: We thank the Reviewer for their time and effort whilst reviewing our manuscript, and for acknowledging the importance in our study.

Areas for Improvement

Methodological Considerations

Comment 1: Validation of AI-based Data Extraction: The study reports an accuracy rate of 72%-95% across various extracted variables. However, further clarification on the error types and specific misclassifications would strengthen the reliability of the findings. Additionally, a more extensive validation process beyond a 1% sample might be warranted to ensure robustness.

Reply 1: We appreciate the Reviewer’s request for further clarification on the types of errors observed in AI-based data extraction. To assess model performance, we conducted a manual validation of 1% of the dataset (n=838), which revealed three primary sources of misclassification. The most common issue was context misinterpretation, where the LLM occasionally misattributed weight loss values when multiple individuals were mentioned in a single post. For example, in posts such as “My friend lost 15 kg, I only lost 5 kg”, the model sometimes assigned the wrong weight loss figure to the primary user. Similarly, when users referenced multiple dieting periods, such as “I lost 5 kg last year and another 10 kg recently”, the model occasionally extracted only one of the reported values instead of the full weight loss history. Another source of misclassification was implicit mentions of weight loss, where the LLM performed well when weight loss was explicitly stated (e.g., “I lost 10 kg”), but sometimes struggled with indirect statements such as “Started at 100 kg, now down to 85 kg”, failing to infer the weight change. Lastly, unit conversion errors arose in cases. Although we applied regular expressions to standardize weight units, some inconsistencies remained in ambiguous cases.
Despite these errors, our analysis indicates that misclassifications did not introduce systematic bias in weight loss estimation.

As also raised by Reviewer’s 1, comment 4, the accuracy trend graph, shown below

shows that classification accuracy stabilized after approximately 200–300 checked posts and remained consistent at ~72%, suggesting that validating additional cases would not significantly alter our accuracy estimates. Furthermore, other extracted variables such as gender, age, height, and physical activity exhibited high classification accuracy (90–95%), reinforcing confidence in the dataset’s reliability for multivariate analysis. To enhance accuracy in future studies, we propose testing larger versions of Llama 3 (e.g., Llama 3-70B) to improve contextual understanding and experimenting with alternative state-of-the-art LLMs such as GPT-4, Claude, or Mistral. We have now clarified these misclassification sources, their impact, and proposed improvements in the Discussion section (p. 8, lines 279-284 and 290-292).

 Comment 2: Handling of Missing Data: Given that complete cases were only 2,416 posts, a discussion on potential biases introduced by missing data and strategies to address them (e.g., imputation methods) would enhance the study’s validity.

Reply 2: We acknowledge that missing data is a limitation in our study, and we now discuss potential biases introduced by incomplete cases. Given that only 2,416 of the ~35,000 posts contained complete information, the proportion of missing data is substantial (~93%), making imputation methods unfeasible. Traditional imputation approaches, such as multiple imputation or machine learning-based imputations, require a sufficiently large proportion of observed data to reliably estimate missing values (please see https://doi.org/10.1186/s12874-017-0442-1). In our case, the extent of missingness would likely introduce additional uncertainty rather than improve data quality. Regardless, results regarding the association of diet duration and weight loss were similar between available and the complete cases, albeit slightly attenuated in the latter case (please see revised Table 3 and p. 6, line 179).

Instead of imputation, we opted for an analysis based on complete cases while acknowledging that missingness could be non-random. Users may have selectively reported certain aspects of their experiences while omitting others, potentially leading to biases in the observed dataset. For example, posts with more detailed weight-related data may come from more engaged or successful individuals, which could contribute to an overestimation of weight loss effects. Conversely, individuals who faced challenges or dropped out of the diet may have been less likely to provide follow-up updates.

To account for this limitation, we have now expanded the discussion (please see p. 8, lines 259-263) to highlight the potential impact of missing data and emphasize that our findings reflect self-reported experiences rather than a comprehensive assessment of all individuals attempting the diet.

Comment 3: Reddit Representativeness: While Reddit provides a rich source of qualitative and quantitative data, users may not be representative of the general population. A more explicit discussion on demographic biases (e.g., age, gender, socio-economic status) and their implications on generalizability is recommended.

Reply 3: Following the Reviewer’s comments and Reviewer’s #1, Comment 11c, we acknowledge in the discussion (please see p. 7, lines 226-242) that representativeness might be an issue when collecting data from online resources. However, the composition of the dataset, including sex distribution, reported health conditions, and behaviors, reflects the nature of self-selected participants in online diet discussions. While this may introduce biases, particularly in engagement patterns and reporting tendencies, our aim was to explore trends in self-reported experiences rather than establish epidemiological prevalence.

Interpretation of Findings

Comment 4: Potential Confounding Factors: The study acknowledges the associations between weight loss and diet adherence, gender, and health conditions. However, the role of other confounders, such as caloric intake, dietary quality, and metabolic adaptations, is not discussed. Additional consideration of these factors would provide a more nuanced interpretation of results.

Reply 4: We appreciate the Reviewer’s suggestion to consider additional confounding factors such as caloric intake, dietary quality, and metabolic adaptations. However, our study aimed to explore self-reported experiences rather than assess the magnitude of effects or the effectiveness of the ketogenic diet. Given this focus, we did not collect data on dietary intake or metabolic factors, as our primary objective was to analyze patterns in user-generated discussions rather than conduct a controlled dietary assessment. Nonetheless, we acknowledge that these factors may influence individual weight loss trajectories and dietary adherence, and future research could further explore these aspects in more structured study designs (please refer to p. 8, lines 251-256).

Comment 5: Comparison to Traditional Dietary Studies: The study could benefit from a comparative analysis between findings from Reddit users and existing clinical or epidemiological studies on the ketogenic diet. This would contextualize the results within broader nutritional research.

Reply 5: We appreciate the Reviewer’s suggestion regarding a comparative analysis with traditional dietary studies. As noted in the discussion (p. 6, lines 190-194), our findings align with established research on the ketogenic diet’s effectiveness in weight loss [Sherer, E.L.; Sherer, J.A. JAAPA 2008, 21, 31–39], particularly for individuals seeking rapid weight reduction or improved glycemic control [Moriconi, E.; et al. Nutrients 202113, 758]. However, given that our primary aim was to explore self-reported experiences through user-generated content, rather than conduct a systematic comparison with clinical or epidemiological studies, an in-depth analysis of this aspect falls beyond the scope of our study. Future research could further investigate how insights from online discussions compare to traditional dietary assessments to enhance understanding of digital health narratives.

Comment 6: Limitations of Self-Reported Data: Self-reported weight loss is inherently prone to recall bias and exaggeration. While the study acknowledges this limitation, discussing potential ways to mitigate these biases (e.g., cross-validation with external sources) would be valuable.

Reply 6: Thank you. We have now added a relevant point in the discussion section (p. 7, lines 233-235; p. 8, lines 253-256).

Computational Considerations

Comment 7: Processing Time and Scalability: The study reports an average processing time of 32 seconds per post. A discussion on the feasibility of scaling this approach to larger datasets or real-time dietary monitoring would be beneficial.

Comment 8: Computational Resources: The reliance on a Dell G15 5530 laptop for model execution raises questions about the potential for leveraging high-performance computing (HPC) or cloud-based solutions to improve efficiency and accuracy.

Reply 7-8: Thank you. We appreciate the reviewer's concern regarding the processing time and computing resources used in our study. The choice of the Dell G15 5530 laptop was primarily driven by practical constraints and accessibility. While this setup was sufficient to process the dataset within a reasonable timeframe, we acknowledge that more powerful computing resources could improve both efficiency and accuracy. We address this limitation in the Discussion section and suggest ways to improve the efficiency of this approach (please refer to p. 9, lines 310-332).

Ethical Considerations

Comment 9: User Privacy and Consent: While Reddit data is publicly available, ethical considerations regarding user consent and data usage should be further elaborated. Explicit discussion on ethical approvals and compliance with social media research guidelines (e.g., respecting user anonymity, ethical data scraping) would strengthen the manuscript.

Reply 9: Thank you for this comment. We have now clarified in the discussion section (p. 9, lines 339-343) that our study adheres to the Institutional guidelines for ethics in research and GDPR regulations, and ethical considerations regarding social media data. We emphasize our commitment to respecting user anonymity, following ethical data scraping practices, and complying with social media research guidelines to ensure responsible data usage.

Comment 10: Transparency and Bias in AI Models: Given concerns about biases in AI-generated outputs, a discussion on model transparency, interpretability, and potential mitigation strategies (e.g., human-in-the-loop validation) would be valuable.

Reply 10: We appreciate the Reviewer’s concern regarding biases in AI-generated outputs and the need for greater transparency and interpretability. We have revised Discussion section to reflect this point (p. 9, lines 352-359).

Minor Revisions

Comment 11: Clarity in Data Presentation: The figures and tables should be revised for clarity. For example, Figure 1 could better visualize the filtering process of the dataset, and Table 2 could provide additional statistical comparisons.

Reply 11: We appreciate the Reviewer’s suggestion to improve the clarity of our data presentation. We have now added a comparison between complete and available data in Table 2, as recommended, and clarified the additional analyses in the statistical analysis section (p. 3, lines 126-129). However, it is unclear to us what further refinements should be done to improve Figure 1. We would be grateful for any further clarification that will help us present the information more effectively.

Comment 12: Grammar and Style: Some minor grammatical inconsistencies and typographical errors are present. A thorough proofreading would improve readability.

Reply 12: We have thoroughly proofread and revised the manuscript to improve wording, clarity and overall readability. We hope that these changes align with the standards of the Journal.

Comment 13: References: Ensure that all citations are properly formatted according to the target journal’s guidelines and that references to AI and dietary research are up-to-date.

Reply 13: We have carefully reviewed all references and ensured that they are properly formatted according to the Journal’s guidelines. Additionally, we have checked existing references and added relevant new ones, where necessary (please see highlighted references in the References section) to ensure the manuscript reflects the most current literature.

Conclusion and Recommendation

Comment: This study presents an innovative and valuable contribution to the field of dietary assessment by utilizing AI-powered methods to analyze self-reported weight loss experiences from Reddit. While the methodology is robust, addressing the aforementioned concerns would significantly strengthen the study’s validity and impact. In particular, more detailed discussions on data validation, potential biases, confounders, computational limitations, and ethical considerations would enhance the manuscript’s rigor.

Recommendation: major revisions required before acceptance.

Reply: We appreciate the Reviewer’s concerns. We believe we have addressed all the comments in the revised version of the manuscript, particularly regarding data validation, potential biases, computational limitations, and ethical considerations. We look forward to your evaluation of the updated manuscript.

Round 2

Reviewer 1 Report

Comments and Suggestions for Authors

The authors have performed an amazingly arduous, fruitful revision of their manuscript and clarified a lot of critical points raised by me. Many of my concerns are resolved. Thanks for that engaged work.

In order to further improve the publication, I recommend the following points, based on my initial five key comments on methodology / results and discussion:

a) First: Is the LLM extraction reliable enough to identify "cases" with full data and to properly extract that data?

--> Please provide a comparison of full, complete, included cases and specifically those non-cases, which were excluded for inaccuracy and checked manually (n=838). This comparison is necessary to outrule a selection bias. As you see, there is a shift in gender ratio from all cases (n=35079) to all complete cases (n=2416). A similar shift could be hidden in inaccurately extracted cases.

The figure on your validation approach is definitely worth publishing; please include it in your manuscript.

And: Could you include the number of validated data sets into the flow chart (for each level, separately), so that one can also see, how many of the final cases were cross-checked?

b) Second: Are these individual cases plausible, when considering baseline weight (if reported) and presumed weight loss? Are there duplicate "cases" which were reported in multiple sub-reddits?

I think, these points are less trivial than currently discussed. If future researchers would use AI for data extraction, these aspects become highly relevant, as it provides bias of over/underestimation of effect sizes, bias of implausible cases (e.g. 20 kg weight loss for a normal weight person) and bias of duplicate cases. Ergo, please clarify by your data, to which extent the data is plausible. Higher average weight loss compared to any weight loss reported in well-documented clinical trials sheds serious doubts on the plausibility of the extracted data, even if they were extracted correctly.

Even if weight loss averages from this reddit cohort match results from other digital cohorts, all of them could share the same types of original bias: third-party interests, fake accounts etc. ...

c) Third: Is the total "cohort" plausible in itself or do proportions of sex, health disorders or behavior indicate a bias of any origin?

Thankfully, you acknowledge the potential limitation by an implausible "cohort structure". However, this could be a general problem of AI, not only reddit. And: If online users report undergoing a certain diet for completely different reasons than reported in clinical studies, there is a bias, that one has to be aware of.

d/e) Fourth/Fifth: There is no way to confirm the adherence to that particular diet, when assuming that the cases are genuine. Even if the persons are real, we don't know, what they actually ate or abstained from. There is no way to confirm the claimed effects and the literature suggests, that the claims are highly exaggerated while ignoring side effects, adherence obstacles and so on.

Data on reddit does not necessarily have to be data on what real persons have perceived (rather than objectively measured). It could be - and in many cases might be - fake data on fake persons suggesting something that could have been perceived (or measured). But: by including fake cases without the chance of validation impairs the data quality of all data points collected, even high quality ones.

Furthermore:

Table 2: The statistical comparison needs to be done between included and excluded cases (not included and total cases).

Fig. 2 still bears some imprecision, as Hashimoto is both endocrine and immunological, psoriatic arthritis is immunological and orthopedic. Also; high BP, LDL or cholesterol are abnormal clinical findings, not diseases. Dyslipidemia and hypertension would be correct terminology.

Table 3: Please present the results for sex, lifestyle and health condition as comparison (Mann-Whitney or T-Test), not as correlation. It is worth knowing the actual difference in weight loss between these comparators.

Discussion: It is an important point, that the accuracy for weight loss - the main variable of interest - could not be improved by extensive manual validation.

Once again: chapeau for the interesting topic and the detailed efforts you put into it.

Author Response

Comments and Suggestions for Authors

Comment: The authors have performed an amazingly arduous, fruitful revision of their manuscript and clarified a lot of critical points raised by me. Many of my concerns are resolved. Thanks for that engaged work.

Reply: We thank the Reviewer for the valuable comments that helped us improve the quality of our manuscript, and for acknowledging the substantial changes we have made in the revised text.

In order to further improve the publication, I recommend the following points, based on my initial five key comments on methodology / results and discussion:

Comment a) First: Is the LLM extraction reliable enough to identify "cases" with full data and to properly extract that data?

--> Please provide a comparison of full, complete, included cases and specifically those non-cases, which were excluded for inaccuracy and checked manually (n=838). This comparison is necessary to outrule a selection bias. As you see, there is a shift in gender ratio from all cases (n=35079) to all complete cases (n=2416). A similar shift could be hidden in inaccurately extracted cases.

The figure on your validation approach is definitely worth publishing; please include it in your manuscript.

And: Could you include the number of validated data sets into the flow chart (for each level, separately), so that one can also see, how many of the final cases were cross-checked?

Reply 1: Thank you for your comments. We have now updated the flowchart (revised Figure 2) to include the number of validated cases at each level, making the selection process more transparent. Additionally, we have introduced a new figure (see new Figure 1 and p. 3, lines 121-126) that presents the classification performance of the LLM across all extracted parameters.

To further address potential selection bias, we have revised Table 2 to compare key characteristics between excluded cases (i.e., those missing additional covariates, n=32,673) and complete cases (n=2,413). This comparison highlights that the most notable difference between these two groups is the gender distribution in reported weight loss. We have now explicitly clarified this point in the Discussion section (p. 9-10, lines 300-305).

Comment b) Second: Are these individual cases plausible, when considering baseline weight (if reported) and presumed weight loss? Are there duplicate "cases" which were reported in multiple sub-reddits?

I think, these points are less trivial than currently discussed. If future researchers would use AI for data extraction, these aspects become highly relevant, as it provides bias of over/underestimation of effect sizes, bias of implausible cases (e.g. 20 kg weight loss for a normal weight person) and bias of duplicate cases. Ergo, please clarify by your data, to which extent the data is plausible. Higher average weight loss compared to any weight loss reported in well-documented clinical trials sheds serious doubts on the plausibility of the extracted data, even if they were extracted correctly.

Even if weight loss averages from this reddit cohort match results from other digital cohorts, all of them could share the same types of original bias: third-party interests, fake accounts etc. ...

Reply 2: We appreciate the Reviewer’s concerns regarding data plausibility and potential biases in weight loss estimates. In our analysis, baseline weight data were not collected, limiting direct assessment of plausibility based on initial weight status. However, to mitigate extreme values, all numeric data were winsorised at 1%, reducing the influence of implausible outliers. We have now clarified this in the Statistical analysis section (p. 4, lines 149-151). Additionally, while duplicate cases across subreddits were not systematically screened, our dataset was restricted to a single subreddit (r/keto), minimizing the risk of widespread duplication across communities. Moreover, of all the posts we extracted, we have identified only 48 duplicates using fuzzy logic, suggesting minimal impact on the overall results. For consistency we have now excluded these from the analysis (pls. see revised flowchart in Figure 2 and Results section). Regardless, we have explicitly mentioned in the Discussion section that in future studies when incorporating data from multiple subreddits, username matching and text similarity algorithms should be used to identify highly similar posts (i.e., posts with near-identical wording across different subreddits) (p. 10-11, lines 354-361).

We agree with the Reviewer that biases such as third-party influences, fake accounts, or selective reporting may still affect the validity of our findings, despite alignment with other digital cohorts in social media analytics. While these risks cannot be fully eliminated, we emphasize in the Discussion section that AI-extracted data should be interpreted with caution and ideally complemented with traditional research methods (see p. 9, lines 248-250, and p. 12, lines 446-449). Regardless, we acknowledge that anonymity helps reducing desirability bias, as users have fewer incentives to misrepresent their weight loss compared to commercial platforms that promote specific diets or supplements (p. 9, lines 251-254).

Comment c) Third: Is the total "cohort" plausible in itself or do proportions of sex, health disorders or behavior indicate a bias of any origin?

Thankfully, you acknowledge the potential limitation by an implausible "cohort structure". However, this could be a general problem of AI, not only reddit. And: If online users report undergoing a certain diet for completely different reasons than reported in clinical studies, there is a bias, that one has to be aware of.

Reply 3: We acknowledge that cohorts extracted from social media platforms like Reddit may not fully align with traditional clinical populations undergoing dietary interventions. However, the demographic characteristics and reported health conditions in our extracted cohort align with trends observed in other digital health studies. We recognize that self-selection bias, differences in motivations for dietary changes, and the potential overrepresentation of certain groups (e.g., younger individuals, those with specific health concerns) may impact generalizability, as discussed in p. 9, lines 256-258. Furthermore, online users may engage in diet-related discussions for diverse reasons beyond those typically documented in clinical studies, introducing a potential selection bias. To address this, we have expanded the Discussion to further contextualize these differences and emphasize the need for cautious interpretation of findings derived from social media cohorts (please see p. 9, lines 262-272). Additionally, we highlight the importance of integrating AI-extracted cohorts with traditional clinical and epidemiological samples to better understand these discrepancies (p.3, lines 112-116; p. 12, lines 425-427; p. 12, lines 446-449).

Comments d/e) Fourth/Fifth: There is no way to confirm the adherence to that particular diet, when assuming that the cases are genuine. Even if the persons are real, we don't know, what they actually ate or abstained from. There is no way to confirm the claimed effects and the literature suggests, that the claims are highly exaggerated while ignoring side effects, adherence obstacles and so on.

Data on reddit does not necessarily have to be data on what real persons have perceived (rather than objectively measured). It could be - and in many cases might be - fake data on fake persons suggesting something that could have been perceived (or measured). But: by including fake cases without the chance of validation impairs the data quality of all data points collected, even high quality ones.

Reply 4-5: We appreciate the Reviewer’s concern regarding the inability to actually confirm users’ dietary adherence and the claimed effects of diets, and the potential presence of fake or exaggerated cases that could compromise data quality. We fully acknowledge that our dataset is based on self-reported Reddit posts, which inherently lack external validation regarding dietary adherence, actual intake, and the claimed effects of different diets. As we mention in the Discussion section (p. 9, lines 268-270) any study relying on self-reported data suffers from the risk of recall bias, selective reporting, and exaggeration, which can influence the accuracy of the extracted information. While our study aimed to analyze naturally occurring diet-related discussions, we recognize that online narratives may not fully reflect real-world dietary behaviors or physiological outcomes. We have clarified this in the Discussion section (p. 9, lines 270-272).

Additionally, we agree that user-generated content on social media platforms carries the risk of misinformation, exaggerated claims, and even fabricated accounts. However, existing research on digital health data suggests that while some posts may be misleading or inauthentic, large-scale trends from such datasets often align with broader public health patterns. To mitigate potential biases, we suggest that future studies could integrate AI-based filtering techniques to detect anomalies, cross-reference findings with validated datasets, or combine social media data with more structured epidemiological sources (p. 9, lines 290-293; p. 10, lines 339-341; p. 11, lines 408-412).

Furthermore:

Comment 6: Table 2: The statistical comparison needs to be done between included and excluded cases (not included and total cases).

Reply 6: Following the Reviewer’s comment we have now updated Table 2 for the comparison between the excluded and the complete cases. Please refer to revised Table 2 and p.5, lines 171-180.

Comment 7: Fig. 2 still bears some imprecision, as Hashimoto is both endocrine and immunological, psoriatic arthritis is immunological and orthopedic. Also; high BP, LDL or cholesterol are abnormal clinical findings, not diseases. Dyslipidemia and hypertension would be correct terminology.

Reply 7: We appreciate the Reviewer’s attention to detail and acknowledge the complexity of disease classification, particularly for conditions with overlapping immunological, endocrine, and musculoskeletal components. According to the ICD-10 classification, autoimmune thyroiditis, including Hashimoto’s thyroiditis, falls under the broader category of “Endocrine, nutritional, and metabolic diseases”. While Hashimoto’s encephalopathy is an autoimmune neurological manifestation, it is a rare condition, and we assumed it was not prevalent within the extracted dataset. Regarding psoriatic arthritis, ICD-10 classifies it as an inflammatory arthropathy under “Diseases of the musculoskeletal system and connective tissue”. Although it has an immunological basis, its primary classification aligns with musculoskeletal disorders. Lastly, we agree with the Reviewer that high blood pressure, LDL, and cholesterol are abnormal clinical findings rather than diseases per se. However, these terms reflect how Reddit users self-reported their health conditions. In response to this concern and to enhance clarity, we have revised the terminology to align with appropriate disease classifications (p. 8, line 193).

Comment 8: Table 3: Please present the results for sex, lifestyle and health condition as comparison (Mann-Whitney or T-Test), not as correlation. It is worth knowing the actual difference in weight loss between these comparators.

Reply 8: We have added a new Table 3 for the comparisons of weight loss between socio-demographic characteristics of the complete cases sample (n=2,413). Please refer to new Table 3 and p. 7, lines 197-203.

Comment 9: Discussion: It is an important point, that the accuracy for weight loss - the main variable of interest - could not be improved by extensive manual validation.

Reply 9: We appreciate the Reviewer’s important observation regarding the limitations of manual validation in improving weight loss extraction accuracy. To explicitly address this point, we have revised the Discussion section (p. 10, lines 334-341) to highlight that extensive manual review did not lead to significant improvements in accuracy and suggest ways to mitigate this bias in future studies.

Once again: chapeau for the interesting topic and the detailed efforts you put into it.

Reply: Thank you for the constructive feedback. We have tried our best to address all the comments raised by the Reviewer and will look forward to your evaluation.

Reviewer 2 Report

Comments and Suggestions for Authors

The authors have responded adequately to all requests and comments. The article could be published in this format.

Author Response

Comment: The authors have responded adequately to all requests and comments. The article could be published in this format.

Reply: We thank the Reviewer for their constructive comments that helped us improve the quality of our work. We are grateful for the positive feedback and for acknowledging that our work is worth publishing.
